# *Akkermansia muciniphila* and HCC: A Gut Feeling

**DOI:** 10.3390/curroncol32100577

**Published:** 2025-10-17

**Authors:** Mario Capasso, Marco Sanduzzi-Zamparelli, Valentina Cossiga, Maria Guarino, Stefania Murzilli, Alessandra Pelagalli, Domenico Sorrentino, Alon Rutigliano, Filomena Morisco

**Affiliations:** 1Diseases of the Liver and Biliary System Unit, Department of Clinical Medicine and Surgery, University of Naples “Federico II”, 80131 Naples, Italy; valentina.cossiga@unina.it (V.C.); maria.guarino@unina.it (M.G.); alongregory.rutigliano@unina.it (A.R.); filomena.morisco@unina.it (F.M.); 2BCLC Group, Institut d’Investigacions Biomèdiques August Pi i Sunyer (IDIBAPS), 08036 Barcelona, Spain; msanduzzi@clinic.cat; 3Liver Oncology Unit, Hospital Clínic, 08036 Barcelona, Spain; 4Centro de Investigación Biomédica en Red en Enfermedades Hepáticas y Digestivas (CIBEREHD), 28029 Madrid, Spain; 5Biofarma S.r.l., 21013 Gallarate, Italy; murzilli.stefania@biofarmagroup.it; 6Department of Advanced Biomedical Sciences, University of Naples “Federico II”, Via Pansini n. 5, 80131 Napoli, Italy; alessandra.pelagalli@unina.it; 7CEINGE-Biotecnologie Avanzate Franco Salvatore, Via G. Salvatore 486, 80145 Napoli, Italy; sorrentinod@ceinge.unina.it; 8Task Force on Microbiome Studies, University of Naples “Federico II”, 80131 Naples, Italy

**Keywords:** *Akkermansia muciniphila*, hepatocellular carcinoma, gut microbiota, immunotherapy, checkpoint inhibitors, tumor microenvironment

## Abstract

**Simple Summary:**

The gut microbiome plays a crucial role in regulating metabolism, inflammation, and immunological pathways. Increasing evidence shows that the microbial ecosystem can also influence many hepatological conditions through the gut–liver axis, shaping disease development, progression, and response to treatment. Akkermansia muciniphila is one of the rising stars of this ecosystem, playing a key role in maintaining intestinal integrity and promoting immune activation. Its presence may also improve the efficacy of antitumoral systemic therapy, including immunotherapy for hepatocellular carcinoma. Although current data come from preclinical and small clinical studies, these insights highlight a promising frontier where microbiome modulation could complement oncological treatments and improve outcomes for patients with liver cancer. In this review, we discuss the biological pathways regulated by the gut microbiome, particularly those involving Akkermansia muciniphila, their relevance in the development of liver cancer, and how they could be linked to the modulation of immunotherapy response.

**Abstract:**

Immune checkpoint inhibitors (ICIs) have radically changed the landscape of systemic treatment for hepatocellular carcinoma (HCC). Recently, there has been increasing interest regarding the relationship between the gut microbiome and the response to immunotherapy in oncological treatments. Among the gut commensals, *Akkermansia* (*A.*) *muciniphila* has gained increasing attention in the literature. *A. muciniphila* may affect the tumor microenvironment and enhance the efficacy of systemic therapies, including ICIs and targeted agents, by shaping host immune responses and metabolic pathways. This narrative review summarizes the current knowledge on *A. muciniphila* and its potential interaction with systemic therapies for HCC, focusing on its immunostimulatory properties, including enhancement of cytotoxic CD8^+^ T-cell activity and reversal of immunosuppressive tumor microenvironments. The therapeutic role of *A. muciniphila* might represent a novel and promising weapon in the HCC field, although the road is still long and the scientific evidence still remains in an exploratory stage. Its integration into clinical practice, however, requires robust clinical trials and a deeper understanding of its interactions within the gut–liver axis and tumor ecosystem.

## 1. Introduction

Hepatocellular carcinoma (HCC) is the sixth most common malignancy worldwide, being the most common primary liver malignancy in adults and the fifth most common cancer [1]. It is one of the most lethal gastrointestinal tumors, with a five-year survival rate of approximately 21% [2]. Its burden is particularly high in regions with a high prevalence of chronic liver diseases [3]. The pathogenesis of HCC is complex and multifactorial, often arising in the context of chronic inflammation, liver fibrosis, and cirrhosis caused by viral hepatitis [4], alcohol abuse, or metabolic disorders, representing an increasing phenomenon attributed to shifts in lifestyle trends [5,6]. Unfortunately, most HCC patients are diagnosed at an advanced stage, in which treatments are based on systemic therapy [3]. As for most cancers, immunotherapy-based treatments have also gained a crucial role in the field of systemic therapy for HCC. However, only a small proportion of patients respond to treatment, with objective response rate (ORR) observed in almost one-third of the patients [7,8,9,10,11,12], while a non-negligible portion experiences treatment-failure. Numerous factors could influence the response to systemic therapy; in the sections below, we will focus on the potential role of the gut microbiome in this context.

## 2. Methods

This narrative review aims to examine the potential role of microbiomes in HCC, particularly of *Akkermansia muciniphila*, integrating the current evidence base with clinical expertise. The literature search was conducted across PubMed, Scopus, Web of Science, Google Scholar, and the Cochrane Library, considering articles published in English up to July 2025 and in accordance with SANRA (Scale of the Assessment of Narrative Review Articles) criteria, which assess the methodological quality across topic significance, clarity of aims, adequacy of the literature search, referencing quality, strength of scientific reasoning, and appropriateness of data presentation [13]. Boolean operators were applied to combine the following search terms: (“*Akkermansia muciniphila*” OR “microbiome” OR “microbiota” OR “gut microbiome” OR “gut microbiota”) AND (“Hepatocellular carcinoma” OR “HCC” OR “liver cancer” OR “liver neoplasm”) AND (“Immunotherapy” OR “ICIs” OR “Immuno-checkpoint inhibitor”. Study selection was performed independently by authors who screened titles, abstracts, and, when necessary, full texts.

## 3. From Liver Disease to HCC: The Role of Gut Microbiome

The role of gut microbiota is growing interest in chronic diseases. Through its metabolites and related molecules, it could interact with various signaling pathways and influence the development and progression of liver diseases, including cancer. This led to the well-known concept of “gut–liver axis”, a two-way pathway in which bacterial components, immune signals, and microbial metabolites could interact between the gut and the liver, influencing both of them [14].

Compared to healthy people, patients with advanced chronic liver disease present dysbiosis [15]. Specifically, cirrhotic patients present a lower rate of butyrate-producing species like *Coprococcus* spp., *Faecalibacterium prausnitzii*, and members of the *Lachnospiraceae* and *Ruminococcaceae*, while an overgrowth of *Veillonella*, *Streptococcus*, and *Clostridium perfringens* occurs [16]. Moreover, a case–control observational study demonstrated distinct pattern in gut microbiota composition and also between cirrhotic patients with and without HCC. Specifically, *Alphaproteobacteria* and *Verrucomicrobia* phylum were significantly reduced in cirrhotic patients with HCC compared to those without HCC [17]. Indeed, patients with liver cirrhosis and HCC show specific patterns of dysbiosis, characterized by the overrepresentation of pro-inflammatory and pathogenic bacteria (such as *Enterobacter*, *Enterococcus*, *Klebsiella*) with a concomitant decreased beneficial bacteria (like *Akkermansia*, *Bifidobacterium*, and butyrate-producing species) [18].

The Cirrhosis Dysbiosis Ratio (CDR), consisting of the ratio of pathogenic to beneficial bacterial taxa, is a suitable parameter to indicate microbiome alterations. As liver damage worsens, CDR falls, and the lower the ratio, the higher the level of serum endotoxins, as well as the Model for End-Stage Liver Disease (MELD) score, and the likelihood of liver-related events, like hepatic encephalopathy, organ failure, and death [19].

### Pathways and Signaling Behind Gut—Liver Relationship

Pro-inflammatory cytokines are the main actors in the pathway related to liver damage progression. Among them, tumor necrosis factor- α (TNF-α) and Interleukin-6 (IL-6) [20] trigger the inflammatory process by the interaction of Toll-like receptors (TLRs) pathway (specifically stimulating TLR4-dependent signaling) with hepatic immune cells, such as hepatic stellate cells (HSCs) and liver-specific mesenchymal cells that contribute to liver physiology, fibrogenesis, and inflammation [21]. The chronicization of this process leads to an immune shift favoring a less cytotoxic phenotype (e.g., Th1-related responses) and a more immunotolerant profile (e.g., regulatory T cells, Tregs) [22] (Table 1). These pathways outline unique microbial signatures, with pro-inflammatory profiles, linked to the development of cirrhosis and HCC in patients with chronic liver disease. In the following sections, we will detail the mechanisms of signaling and how the gut microbiome fits into this context.

Dysbiosis covers the first step of this process, leading to the condition known as “leaky gut,” characterized by increased intestinal permeability that facilitates the translocation of microbial products. Through the portal vein, the liver is exposed to gut-derived microbial metabolites, bile acids, short-chain fatty acids (SCFAs) and pathogen-associated molecular patterns (PAMPs). In healthy people, a balanced gut microbiota contributes to maintaining system tolerance and supports normal liver metabolism [23]. This condition could promote the microbial-associated molecular pattern (MAMP) translocation, including lipopolysaccharide (LPS) and lipoteichoic acid (LTA), which can activate TLR2 and TLR4, regarded as key players of chronic inflammation, fibrogenesis, and tumorigenesis [24,25,26]. Accordingly, previous preclinical studies showed that the intestinal microbiota and TLR4 activation promote liver tumor growth in mice exposed to diethylnitrosamine (DEN) and carbon tetrachloride (CCl_4_), through the expansion of monocytic myeloid-derived suppressor cells (mMDSCs), and reduced CD8^+^ T-cell abundance, indicating suppression of antitumor immunity [27,28].

Bile acids (BAs), whose synthesis is closely linked to gut microbiota metabolism, may also have a significant impact in hepatocarcinogenesis, as well as cholangiocarcinogenesis [29]. In particular, increased conversion of primary to secondary bile acids by an altered gut microbiota has been associated with the downregulation of the cytokine C-X-C Motif Chemokine Ligand 16 (CXCL16) in natural killer T cells (NKTs) and this has been proposed to manage a weaker antitumor immune response [30]. Moreover, dysbiotic status causes increased levels of Deoxycholic acid (DCA), synthetized by bacteria that promote hepatocarcinogenesis, while this condition has been observed to increase TLR2 in HSCs that resulted in the tumor-promoting senescence-associated phenotype [24]. Dysbiosis-induced BA imbalance (increased Deoxycholic/Lithocholic acid, DCA/LCA ratio) drives activation of oncogenic pathways, JAK-STAT3, NF-κB, promoting hepatocyte proliferation and M2 macrophage polarization [31]. Another underlying mechanism proposed is self-induced: elevated bile acid levels further activate the transcriptional cofactor YAP with a consequently negative feedback on Farnesoid X receptor (FXR), a feedforward cycle supporting inflammation and leading to HCC development [32,33]. Likewise, metabolic disorders are responsible for an altered gut microbiota, leading to elevated levels of DCA, as demonstrated by Yoshimoto, S. et al. [34]. DCA is a secondary bile acid produced by Gram-positive bacteria and, when accumulated, has the potential to induce DNA damage by promoting positive feedback on HSCs. DCA-related senescence phenotypes link gut microbiota dysbiosis to liver cancer progression [35].

Lastly, the ability of the liver to regenerate after injury or surgical resection is also influenced by the gut microbiota. Hepatocytes, Kupffer cells, and HSCs can all be impacted by changes in bile acid profiles and microbial metabolites, which can alter regenerative responses. In particular, Hu Y. et al. [36] demonstrated that specific gut microbial patterns enhanced liver regeneration in C57Bl/6J murine models of dysmetabolic liver disease. In this context, *Akkermansia muciniphila* (*A. muciniphila*)-based treatment exhibited a reduced liver lipid accumulation, promoting liver regeneration, suggesting potential therapeutic approaches through the manipulation of gut microbiota to improve outcomes in liver diseases. The number of scientific publications focusing on specific bacterial strains, such as *A. muciniphila* and *Bifidobacteria*, has increased in recent years. In particular, *A. muciniphila* appears to play a promising role in the gut–liver axis, potentially affecting both the development and progression of chronic liver disease and HCC, and its role will be discussed in the paragraphs below.

## 4. *Akkermansia muciniphila*: The Rising Star of Gut Microbiota

*Akkermansia muciniphila*, a Gram-negative, anaerobic, mucin-degrading bacterium, belonging to the phylum *Verrucomicrobia*, has gained a growing interest due to its peculiar characteristics in modulating metabolic homeostasis and immune responses. The whole signaling pathway and biological effects of *A. muciniphila* are the subject of numerous studies, despite the lack of a deep understanding about mechanisms underlying its biological functions. The most relevant pathways are summarized in Figure 1.

The abundance of *A. muciniphila* is likely to correlate with goblet cells and mucin production, as well as its immunomodulation function, through IFN-γ, IL-6, and TNF-α induction, and a reduction in immunosuppressive regulatory T cells (T_reg_). Accordingly, *A. muciniphila* administration was shown to drive T-cell differentiation in germ-free mice, enhancing a cytotoxic immunological profile [37,38]. As a mucin-degrading bacteria, *A. muciniphila* contributes to both qualitative and quantitative maintenance of the intestinal mucus layer integrity, playing a crucial role in tightening epithelial junctions and limiting microbial translocation [39]. This function is particularly important due to its improving effect on the potentially harmful consequences of leaky gut [40].

Across chronic liver disease, as already discussed, an inverse relationship between *A. muciniphila* and cirrhosis is well established [16,17,18], and its administration has been shown to improve liver fibrosis in a murine model [41]. Furthermore, evidence regarding the association between viral or non-metabolic etiologies and *A. muciniphila* abundance is still lacking. In HBV patients, *A. muciniphila* appears enriched during immune-active phases, while it is depleted in immune-tolerant states. Moreover, *A. muciniphila* shows a positive correlation with CD3^+^ T, CD4^+^ T, and CD8^+^ T-cell counts, supporting its immunomodulatory role, while being negatively correlated with liver dysfunction [42]. Interestingly, in non-treated chronic HCV-related hepatitis patients, *A. muciniphila* shows high abundance, which may act as a compensatory mechanism to counteract chronic inflammation, supporting gut health [43]. However, stronger evidence to date mainly concerns its metabolic implications.

Several studies investigated the involvement of such a gut bacterium in metabolic disorders. *A. muciniphila* abundance undergoes a significative reduction in obesity, insulin resistance, type II diabetes, dysmetabolic-related liver disease and inflammatory bowel disease, suggesting its broad impact on metabolic health [44,45]. The modulation of the gut–liver axis by enhancing gut barrier performance and immune responses is the key point related to *A. muciniphila*. In fact, this represents the mechanism through which it could improve hepatic steatosis and inflammation [46]. Furthermore, its administration in both murine models and humans has been shown to improve Metabolic Dysfunction-Associated Steatotic Liver Disease (MASLD) severity and reduce HCC risk via CXCR6+ NKT cell activation [47]. It has also been demonstrated that administration of *A. muciniphila* reduces hepatic cytolysis biomarkers, including aminotransferases and gamma-glutamyl transpeptidase, reducing pro-inflammatory levels of both IL-2 and IFN-γ in dysmetabolic-related liver disease [48,49].

Specific microbial components have been studied to better understand their antitumor activity. In particular, the outer membrane protein Amuc_1100, the aspartic protease Amuc_1434, and the acetyltransferase Amuc_2172, were demonstrated to modulate immune cell activity, induce apoptosis in tumor cells, and strengthen the intestinal barrier. *Akkermansia*-derived membrane protein Amuc_1100 has been shown to downregulate NOD-like receptor thermal protein domain associated protein 3 (NLRP3)-based inflammasome pathways, as well as TLR4/NF-κB signaling in both hepatic and intestinal tissues, improving cytotoxic immune response in a cancer model [45]. Amuc_1434, whose role has been mainly studied in colorectal cancer cells, upregulated p53, the key protein of the cell cycle, and apoptosis processes and played a role as a trigger of the death receptor pathway and mitochondrial apoptosis by upregulating tumor-necrosis-factor-related apoptosis-inducing ligand (TRAIL), indicating activation of the TRAIL-mediated apoptosis pathway. It was also the first identified *A. muciniphila*-derived protease that can degrade the mucin Muc2 and induce apoptosis in colorectal cancer cells via a TRAIL-dependent pathway [50]. Amuc_2172 enhanced CD8^+^ cytotoxic activity by indirectly promoting transcription and secretion of heat-shock protein 70 (HSP70) in cancer cells [51].

Finally, *A. muciniphila* reshaped bile acid pool. Indeed, it could reduce secondary bile acid (DCA, LCA) synthesis and promote the intestinal FXR–FGF15 axis, a pathway that plays an important role in the regulation of host metabolism, potentially inhibiting progression from MASLD to HCC [52]. By reducing these BAs, *A. muciniphila* is involved in immunological anti-tumorigenesis pathway. BA metabolism is also managed by Amuc proteins. Specifically, Amuc_1100 upregulated bile acid receptors TGR5 and FXR in the liver and ileum, downregulated BA synthesis enzymes (CYP7A1, CYP8B1) and transporters (ASBT), suggesting reduced hepatic BA synthesis and absorption, and upregulated alternative pathway enzyme (CYP7B1). These mechanisms strengthen the ability of *A. muciniphila* in enhancing tumor growth control [53].

These biological properties represent the rationale that allowed *A. muciniphila* to have an impact on the development and, as will be discussed in the following section, treatment response of HCC by regulating systemic inflammation and tumor immunological microenvironment.

## 5. Immunotherapy in HCC

Recently, ICIs have landed in the field of first-line therapy for advanced HCC. Indeed, after tyrosine kinase inhibitors “era”, the landscape of systemic treatment has rapidly evolved, highlighting the importance of ICIs to selectively target programmed cell death protein 1 (PD-1), its ligand PD-L1, and cytotoxic T-lymphocyte antigen-4 (CTLA-4), becoming key components of therapeutic strategies [54,55].

The milestone was the approval of the combination of atezolizumab (anti–PD-L1) and bevacizumab (anti-VEGF), which demonstrated a significant improvement in both overall survival (OS) and progression-free survival (PFS) compared to sorafenib, the previous standard of care [8,56]. Later, other ICI-based regimens have been shown to improve survival in patients [7,8,9,10,11,12].

Nevertheless, although significant advances in immunotherapy led to highly effective therapeutic responses, particularly using combination strategies that overcome single-agent ICI-based regimens, only a portion of patients respond to these therapies, and resistance mechanisms might be associated with, although not completely exhaustive, several factors [57], such as (1) low or absent expression of immune checkpoints, which may result in target loss and therapeutic resistance; (2) infiltration of immunosuppressive cells and higher levels of inhibitory cytokines (e.g., TGF-β, IL-10, and IDO), which can suppress antitumor immunity and facilitate immune escape; (3) the presence of an immunosuppressive tumor microenvironment, including tumor-associated macrophages (TAMs), Kupffer cells, and MDSCs, that may promote the expansion of Tregs; and finally, (4) alterations in the gut microbiota, which are increasingly recognized as important modulators of immune responses and ICI efficacy [58]. Indeed, clinical studies have identified gut dysbiosis markers and fecal calprotectin as predictors of response to ICIs in HCC, suggesting that microbiota profiling could aid in tailoring personalized treatment approaches [59]. Further studies aimed at better addressing these issues are mandatory.

## 6. *Akkermansia muciniphila* and Immunotherapy in HCC

The gut–liver axis and the immune microenvironment have been shown to represent two sides of the same coin in the field of immunotherapy response to HCC. Specific microbial compositions, including *A. muciniphila*, *Bifidobacterium*, and *Bacteroides fragilis* are associated with improved response to anti-PD-1/PD-L1 and anti-CTLA-4-based treatments [60,61].

Among the most studied bacteria in this field, *A. muciniphila*, plays a crucial role in the maintenance of the intestinal barrier and immune system modulation [37]. This growing interest is largely driven by the evidence that oral supplementation is safe and may improve metabolic and neoplastic diseases, also improving anti-PD-1 therapy efficacy.

Indeed, as well described by Wu, X.Q. et al. [62], a higher abundance of *A. muciniphila* is correlated with a reduced incidence of MASLD-HCC and oral administration of *A. muciniphila* during anti-PD-1-based therapy in HCC mice which led to maximal tumor suppression, increased T-cell infiltration/activation, while its depletion is consistently associated with a poor outcomes. This highlights the therapeutic potential of *A. muciniphila* in combination with ICI-based therapies. Moreover, in another murine model using C57BL/6J mice with Hepa1-6 tumor xenografts as an HCC model [63] treatment with anti-PD-1 alone inhibited tumor growth; however, concurrent administration of *A. muciniphila* further and significantly reduced both tumor volume and weight (*p* < 0.05).

Finally, a small prospective study conducted on eight advanced HCC patients demonstrated that response to anti-PD-1 therapy was associated with the presence of *A. muciniphila* and various *Ruminococcaceae* species in stool samples, while non-responder patients exhibited increased microbial dissimilarity [64].

Multiple pathways are used by *A. muciniphila* to enhance immunotherapy response and improve the immune system. The mechanisms employed to stimulate the immune microenvironment in HCC involve an increased proportion of CD8^+^ T cells and IFN-γ, IL-12–dependent recruitment of CCR9+ CXCR3+ CD4^+^ T cells into tumors, and the promotion of apoptosis, the same mechanisms investigated in other neoplasms [65]. Metabolomic analysis revealed a significant increase in taurine-conjugated bile acids, including tauroursodeoxycholic acid (TUDCA), taurohyodeoxycholic acid (TDCA), and taurochenodeoxycholic acid (TCDCA), suggesting a protective role mediated by this pathway. High levels of *A. muciniphila*-dependent TUDCA have been associated with enhanced dendritic cell function and T-cell responses, thereby improving tumor immune surveillance [63].

As explained above, bile acid balance can modulate HCC development through the activation of the transcriptional cofactor YAP. This could be one of different pathways driven by *A. muciniphila* to influence HCC development and ICIs-response, as Zhang, Z. et al. demonstrated. Indeed, in the DEN/TCPOBOP-induced liver cancer model [66], the treatment with dihydroartemisinin (DHA), induced YAP1 inhibition, increased *A. muciniphila* abundance, and restored bile acid homeostasis, pointing towards a proportionally direct association between DHA and *A. muciniphila*. These effects may also sensitize HCC to anti-PD-1 immunotherapy. Indeed, the combination of DHA with anti-PD-1 significantly reduced tumor burden compared to ICI alone. These findings support the concept that targeting YAP1 or supplementing *A. muciniphila* may boost anti-PD-1 immunotherapy efficacy in HCC by modifying the gut–liver axis and immune landscape. The YAP1-related pathway is also involved in lipid droplet accumulation within tumor cells that can promote tumor growth [67]. While this highlights yet another strategy employed by cancer cells to support their own survival and proliferation, it also presents a potential therapeutic target.

Finally, despite the poor number of clinical trials regarding the relationship between *A. muciniphila* and HCC, ongoing phase I studies are exploring the safety and feasibility of *A. muciniphila* probiotics in combination with anti-PD-1 monoclonal antibodies in patients with advanced colorectal cancer (NCT06865521). The results of this and other “in recruitment” trials (NCT05865730) are expected to timely provide evidence supporting the strong synergistic effect with immunotherapy. In the future, we will also benefit from the knowledge gained through ongoing clinical trials evaluating the safety and efficacy of Fecal Microbiota Transplantation (FMT) in this setting. Its application potentially restores a healthy gut microbiome, including the re-establishment of beneficial bacteria and influencing the abundance of *A. muciniphila* in dysmetabolic patients [68]. FMT application would be considered as an ancillary treatment to immunotherapy to overcome the limitations posed by resistance [69]. However, in this setting, the lack of standardized protocols and uncertain persistence of transplanted taxa represent additional limitations. Consequently, while FMT may serve as an ancillary strategy to overcome immunotherapy resistance, the current evidence is preliminary.

## 7. Conclusions

In conclusion, in the field of HCC systemic therapy, the gut microbiota modulation is highlighted as a promising approach to enhance immunotherapy efficacy and, regarding the response to ICIs, microbiota composition has been demonstrated to serve as a biomarker in success or failure prediction.

*A. muciniphila* stands out as a promising candidate against cancer development and progression. Moreover, it has demonstrated a strong immune activity and an immune microenvironment sensitizing effect. Despite its promising role, the current body of evidence is still limited by several methodological constraints. Most studies linking *A. muciniphila* to HCC or immunotherapy are derived from small observational cohorts or preclinical murine models, which restrict generalizability. The heterogeneity of animal models (chemical-induced, xenograft, or metabolic-associated HCC) further complicates extrapolation to human disease. In addition, microbiome profiling lacks standardization across studies, with variability in sequencing platforms, bioinformatic pipelines, and taxonomic resolution that may contribute to these limitations.

A deeper understanding of microbiota–immune interactions may shed more light on the innovative therapeutic strategies in HCC, potentially improving patient outcomes by enhancing responses to immunotherapy. Future clinical trials are needed in this field, aiming to explore microbiota-targeted interventions, including *A. muciniphila* supplementation and diet-based modulation to improve outcomes and overcome treatment resistance.

## Figures and Tables

**Figure 1 curroncol-32-00577-f001:**
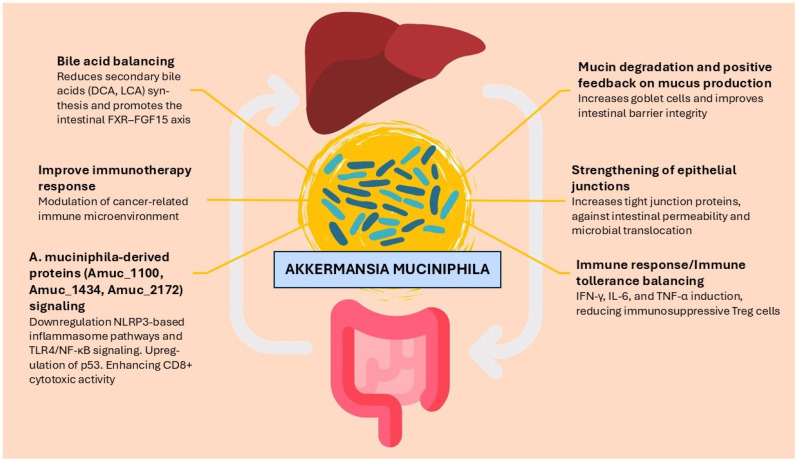
Most relevant pathway in the interaction between *A. muciniphila* and host. Abbreviation: IFN-γ: Interferon-gamma; IL-6: Interleukin-6; TNF-α: tumor necrosis factor-alpha; NLRP3: NOD-like receptor family pyrin domain containing 3; TLR4: Toll-like receptor 4; NF-κB: nuclear factor kappa-light-chain-enhancer of activated B cells; DCA: deoxycholic acid; LCA: lithocholic acid; FXR: Farnesoid X receptor; FGF15: fibroblast growth factor 15.

**Table 1 curroncol-32-00577-t001:** **Most relevant pathway in the interaction between gut microbiota and liver disease.** Dysbiosis increase intestinal permeability due to “leaky gut”. Many pathways are involved in the gut–liver relationship and gut dysbiosis could increase inflammation and disease progression.

Biological Effect	Molecular Signaling
Microbial product translocation	MAMPs, LPS, and subsequent TLR-2 and -4 signaling
Pro-inflammatory cytokines	TNFα, IL-6, and TLR-2 and -4 signaling
Liver HCS activation	Fibrogenesis and tumor-promoting senescence phenotype
Immunotolerant profile	Th1/Treg imbalance: less cytotoxic in a suppressed immune microenvironment
Bile acid imbalance	CXCL16 downregulation in NKT, TLR-2 signaling, YAP activation and FXR negative feedback: pro-inflammatory status

Abbreviations: MAMPs: microbic-associated molecular patterns; LPS: lipopolysaccharide; TLR: Toll-like Receptor; TNF-α: tumor necrosis factor-alpha; IL-6: Interleukin-6; HCSs: hepatic stellate cells; Th1: T helper 1 (subset of CD4^+^ T cells); Treg: regulatory T cells; CXCL16: C-X-C motif chemokine ligand 16; NKT: natural killer T cells; FXR: Farnesoid X Receptor.

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
