# Peer review of "Akkermansia muciniphila and HCC: A Gut Feeling"

_curroncol, 2025, doi:10.3390/curroncol32100577_

Round 1
Reviewer 1 Report
Comments and Suggestions for Authors
I would like to congratulate the authors for performing this narrative review. The manuscript presents a comprehensive narrative review regarding the potential role of Akkermansia muciniphila in hepatocellular carcinoma (HCC). It focuses on the bacterium's immunomodulatory properties, its interactions with the gut-liver axis, and its influence on the efficacy of immune checkpoint inhibitors (ICIs). This subject is of significant relevance, particularly in light of the growing interest in the relationship between the microbiome and cancer, as well as the pressing need for strategies to improve outcomes in HCC immunotherapy.
Althoug I would like to provide some suggestions and comments.
1.-I recommend adding a brief section about the methodology, specifically regarding the literature search and the process used to conduct it. This addition would enhance the quality of the review. Including the SANRA recommendations would be beneficial as part of the methodology section.
2.- Also, be specific within abstract and the manuscript that this is a narrative review.
3.- The manuscript presents a descriptive overview rather than a critical analysis. While it effectively summarizes the existing evidence, it does not sufficiently address the limitations of the cited studies, including small clinical sample sizes, heterogeneity among animal models, and variations in microbiome sequencing methods. It would be beneficial to place greater emphasis on study design and potential biases.
4.-Statements like “A. muciniphila might represent a novel and promising therapeutic weapon in HCC” are premature, considering the limited clinical evidence available. It would be wise to moderate conclusions and highlight that this research is still in the exploratory stage.
Author Response
COMMENT 1: I recommend adding a brief section about the methodology, specifically regarding the literature search and the process used to conduct it. This addition would enhance the quality of the review. Including the SANRA recommendations would be beneficial as part of the methodology section.
RESPONSE 1: Thank you, we have integrated the manuscript by adding a methods section in which we also discussed the criteria for selecting the references.
COMMENT 2: Also, be specific within abstract and the manuscript that this is a narrative review.
RESPONSE 2: Done. Thanks
COMMENT 3: The manuscript presents a descriptive overview rather than a critical analysis. While it effectively summarizes the existing evidence, it does not sufficiently address the limitations of the cited studies, including small clinical sample sizes, heterogeneity among animal models, and variations in microbiome sequencing methods. It would be beneficial to place greater emphasis on study design and potential biases.
RESPONSE 3: We thank the reviewer for this valuable observation. In response, we have revised the Conclusion section to provide a more critical appraisal of the cited studies, highlighting limitations such as small sample sizes, heterogeneity of animal models, and variability in microbiome sequencing methods.
COMMENT 4: Statements like “A. muciniphila might represent a novel and promising therapeutic weapon in HCC” are premature, considering the limited clinical evidence available. It would be wise to moderate conclusions and highlight that this research is still in the exploratory stage.
RESPONSE 4: Thank you, we have emphasized the early and still preliminary nature of the scientific evidence, also integrating “conclusion” with limitations of studies.
Reviewer 2 Report
Comments and Suggestions for Authors
Authors reviewed about dysbiosis and HCC, especially focused on A. muciniphilia. As authors described, efficacy of systemic immunotherapy with ICIs for HCC is limited. Therefore, this review is interesting and informative one. Several issues remained to be addresed.
- The contribution of A. municiphilia to liver disease or HCC is unclear. In section 3, it should be described more detail. The frequency or distribution of A. municiphilia in liver diseases such as viral hepatitis, cirrhosis or HCC should be described.
- In section 5, FMT with A. municiphilia, its possibility or limitation, should be more discussed.
- Some abbreviations should be shown with full spelling (e.g. ORR (section), TNF, IL-6, Treg, CXCL, DCA, LCA (section 2.1), MASLD (section 3), FMT (section 5).
Author Response
COMMENT 1: The contribution of A. municiphilia to liver disease or HCC is unclear. In section 3, it should be described more detail. The frequency or distribution of A. municiphilia in liver diseases such as viral hepatitis, cirrhosis or HCC should be described.
RESPONSE 1: Thank you for your comment. The manuscript already provides an extensive discussion on the role of Akkermansia muciniphila in patients with HCC and its relationship with immunotherapy. In the revised version, we have now integrated a short paragraph addressing its role in cirrhosis and viral liver diseases. Although the available evidence in the literature remains limited, we believe this addition enriches and strengthens the overall perspective of our work.
COMMENT 2: In section 5, FMT with A. municiphilia, its possibility or limitation, should be more discussed.
RESPONSE 2: We thank the reviewer for this important suggestion. We have briefly now discuss the feasibility and limitations of FMT with A. muciniphila, also adding more information about that.
COMMENT 3: Some abbreviations should be shown with full spelling (e.g. ORR (section), TNF, IL-6, Treg, CXCL, DCA, LCA (section 2.1), MASLD (section 3), FMT (section 5).
RESPONSE 3: Done. Thanks
Round 2
Reviewer 1 Report
Comments and Suggestions for Authors
I want to congratulate the authors once again on this manuscript. The authors have responded to all my comments so I have no further comments.
Reviewer 2 Report
Comments and Suggestions for Authors
Revised manuscript was well-addressed to the reviewer's comments or suggestions, and well-written. This study is interesting and informative.